# Alum/CpG Adjuvanted Inactivated COVID-19 Vaccine with Protective Efficacy against SARS-CoV-2 and Variants

**DOI:** 10.3390/vaccines10081208

**Published:** 2022-07-29

**Authors:** Yuntao Zhang, Xiaotong Zheng, Wang Sheng, Hongyang Liang, Yuxiu Zhao, Xiujuan Zhu, Rong Yang, Yadan Zhang, Xiaofei Dong, Weidong Li, Fei Pei, Ling Ding, Zhen Chang, Li Deng, Guangying Yuan, Zhaona Yang, Di Zhu, Xiaoming Yang, Hui Wang

**Affiliations:** 1Beijing Institute of Biological Products Company Limited, Beijing 100176, China; zhangyuntao@sinopharm.com (Y.Z.); zhengxiaotong20@sina.com (X.Z.); ayangyang@163.com (H.L.); zhao10306@126.com (Y.Z.); zhuxiujuan1@sinopharm.com (X.Z.); yangrong0419@126.com (R.Y.); yadan1220@126.com (Y.Z.); dongxiaofei73@cau.edu.cn (X.D.); lwd12378@163.com (W.L.); peifei_pumc@163.com (F.P.); dinglingmail@163.com (L.D.); czhen9008@163.com (Z.C.); drlideng@163.com (L.D.); fy02832@163.com (G.Y.); yangzhaona2020@163.com (Z.Y.); judy_bio@163.com (D.Z.); 2China National Biotec Group Company Limited, Beijing 100024, China; 3College of Life Sciences and Biotechnology, Beijing University of Technology, Beijing 100021, China; shengwang@bjut.edu.cn

**Keywords:** CpG, SARS-CoV-2, Omicron, inactivated vaccine, immunogenicity, safety

## Abstract

Since the beginning of the COVID-19 pandemic, numerous variants of severe acute respiratory syndrome coronavirus 2 (SARS-CoV-2) have emerged, including five variants of concern (VOC) strains listed by the WHO: Alpha, Beta, Gamma, Delta and Omicron. Extensive studies have shown that most of these VOC strains, especially the currently dominant variant Omicron, can escape the host immune response induced by existing COVID-19 vaccines to different extents, which poses considerable risk to the health of human beings around the world. In the present study, we developed a vaccine based on inactivated SARS-CoV-2 and an adjuvant consisting of aluminum hydroxide (alum) and CpG. The immunogenicity and safety of the vaccine were investigated in rats. The candidate vaccine elicited high titers of SARS-CoV-2-spike-specific IgG antibody and neutralizing antibody in immunized rats, which not only neutralize the original SARS-CoV-2, but also showed great cross-neutralization activity against the Beta, Delta and Omicron variants.

## 1. Introduction

Coronavirus disease 2019 (COVID-19), caused by the severe acute respiratory syndrome coronavirus 2 (SARS-CoV-2), was first reported in December 2019 and has become an explosive global pandemic [1]. The symptoms of infected people with this novel coronavirus can range from being asymptomatic or having minimal respiratory symptoms to febrile illness, as well as severe respiratory failure needing ventilator support. The unprecedented pandemic has caused huge economic and social upheaval internationally [2,3,4].

SARS-CoV-2 belongs to subgenus Sarbecovirus of genus Beta coronavirus of the family Coronaviridae. SARS-CoV-2 utilizes angiotensin-converting enzyme 2 (ACE2) as its primary receptor for its entry into human cells [5,6]. The virus contains several structural proteins, one of which is the spike protein (S), a type I fusion transmembrane protein. After extensive glycosylation during the synthesis process, S protein is cleaved into two fragments, S1 and S2. The S1 fragment contains a receptor binding domain (RBD) that can interact with ACE2, whereas the S2 fragment plays a key role in mediating virus integration into the host cell [7]. These features make S protein and RBD the main targets of the host neutralizing antibodies (NAbs) [8]. However, variants have been emerging since the beginning of COVID-19 pandemic, and mutations in the RBD have been detected regularly, which are used to define the numerous variants, including Alpha (B.1.1.7), Beta (B.1.351), Gamma (P.1), Delta (B. 1.617.2) and Omicron (B.1.1.529). Mutations in the RBD directly lead to the enhancement of viral ability of immune escape, such as K417N, E484K, N501Y in Beta (B.1.351) and L452R, T478K in Delta (B.1.617.2) [9]. Surprisingly, the most prevalent variant, Omicron (B.1.1.529), has up to 15 mutation sites in the RBD, most of which are all located near the ACE2 and RBD binding interface, except for G339D, S371L, S373P and S375F [10]. Multiple mutant variants not only increase the contagiousness and pathogenic potential of the virus, but also reduce the neutralizing capacity of vaccine-induced antibodies [11]. As such, SARS-CoV-2 has posed a major challenge to global public health management.

Contemporary COVID-19 vaccine strategies have been deployed globally, because COVID-19 can be controlled with effective vaccines [12,13]. The existing vaccines, including mRNA vaccines, adenovector-based vaccines and inactivated vaccines, have been shown to be effective against SARS-CoV-2 [3]. More importantly, immunization is directly associated with the prevention of viral infection, reduction in viral transmission and attenuation of severe symptoms [14,15]. However, it has been reported that the efficacy, especially the neutralization ability against the variants, wanes along with the decrease in antibody titer after vaccination [16]. Thus, vaccine boosters and further developments of existing vaccines would help prevent the transmission of variants in a shorter period of time [17,18,19]. It is known that fewer uncommon although serious adverse events are observed after homologous or heterologous vaccination immunized with inactivated vaccines [20,21], which means that the inactivated vaccines cover more of the population with underlying diseases compared with mRNA and adenovector-based vaccines [22,23]. In order to take advantage of vaccines and deal with COVID-19 variants, we have modified existing vaccines to increase their resistance to the original strain as well as mutants.

During inactivated vaccine formulation, adjuvants are often added to improve immune responses. Alum, the most commonly used adjuvant, exerts a repository effect, following improved antigen uptake; it also facilitates antigen presentation. Alum is safe and well tolerated, but its adjuvanticity is relatively weak, and it is viewed as an adjuvant which favors Th2 immunity [24,25]. In order to overcome the shortcomings of aluminum-hydroxide-based adjuvants and to evoke more potent immune responses, researchers have explored the development of composite adjuvant vaccines [26,27]. There is now a new synthetic oligonucleotide adjuvant containing an immunostimulatory CpG motif (CpG ODN). It binds to Toll-like receptor 9 (TLR9) and can strongly stimulate the excitation of B cells and plasmacytoid dendritic cells (pDCs). In addition, CpG ODNs also indirectly activate many other types of immune cells through cytokines [28]. CpG ODNs are divided into three classes (class A, B and C) [29]. They are all effective vaccine adjuvants, especially class B is particularly effective at activating B cells [30].

In this study, we developed an inactivated vaccine with a novel adjuvant consisting of CpG 7909, aluminum hydroxide and inactivated SARS-CoV-2. We evaluated the safety and potent adjuvant effects of the class B CpG—CpG 7909 and verified the effectiveness of this vaccine by detecting the neutralization ability of rat serum antibodies against SARS-CoV-2 and a variety of mutants.

## 2. Materials and Methods

### 2.1. Ethics Statement

All the animals involved in this study were housed and cared for in an Association for the Assessment and Accreditation of Laboratory Animal Care (AAALAC)-accredited facility. All the experimental procedures with rats were conducted according to Chinese animal use guidelines and were approved by the Institutional Animal Care and Use Committee (IACUC), IACUC approval number: ACU21-2409. All the animals were anesthetized with isoflurane.

### 2.2. Animal Models

The 6–7 week-old male and female Sprague Dawley rats (207–250 g and 177–218 g, respectively) (SD rats) were obtained from Zhejiang Vital River Laboratory Animal echnology Co., Ltd., and bred and maintained in a specific-pathogen-free (SPF) environment. All animals were allowed free access to water and diet and provided with a 12 h light/dark cycle (temperature: 18–28 °C, humidity: 40–70%).

### 2.3. Vaccine Preparation

SARS-CoV-2 was provided by the Chinese Center for Disease Control and Prevention. Viruses were cultured at a temperature of 36 ± 1 °C and harvested at 72 h after inoculation; then, they were inactivated with β-propiolactone at 2–8 °C for 24 h, followed by chromatography purification [31]. The CpG 7909 (5′-TCGTCGTTTTGTCGTTTTGTCGTT-3′) (abbreviated as CpG) used in this study was synthesized by Sangon Biotech Company (Shanghai, China) and was diluted with double-distilled water and vortexed until completely dissolved. The vaccine was prepared by adding aluminum hydroxide and CpG as the dual adjuvants and diluting with PBS. The final concentrations of aluminum hydroxide and CpG were 0.45 mg/mL and 20 μg/mL, respectively. The inactivated SARS-CoV-2 concentration was 13 U/mL. One dose was 0.5 mL.

### 2.4. Vaccine Immunogenicity Analysis

SD rats were randomly divided into three groups (5 male and 5 female rats in each group), which received four intramuscular injections at two week intervals. The group animals were immunized with physiological saline (as negative control), low-dose vaccine (1 dose/rat) or high-dose vaccine (3 doses/rat), respectively. Sera were collected for specific antibody assays and neutralizing antibody assays to assess vaccine immunogenicity.

### 2.5. Enzyme-Linked Immunosorbent Assay (ELISA)

SARS-CoV-2 S protein antibody titers of serum samples collected from immunized rats were determined by indirect ELISAs: 96-well microtiter plates were pre-coated with S protein at 2–8 °C overnight, and blocked with 2% BSA for 1 h at 25 °C. Sera were tested at a starting dilution of 1:100 and was applied to wells for 2 h at 25 °C, followed by incubation with rabbit anti-Rat IgG (whole molecule)-peroxidase antibody for 1 h at 25 °C after six washes. The plate was developed using TMB, following the addition of 2 M H_2_SO_4_ to stop the reaction, and read at 450 nm with the ELISA plate reader for final data. The average OD values were multiplied by a 1:100 dilution of the serum of the negative control group in the same period by 2.1 as the cut-off value. The antibody titers and the geometric mean titers (GMTs) were statistically analyzed using GraphPad prism program (GraphPad Software, San Diego, CA, USA).

### 2.6. Microneutralization Assay

Rat serum was used to assess the neutralization of SARS-CoV-2 and variants. All procedures were performed in a biosafety level 3 (BSL-3) facility. Briefly, after the inactivation of rat serum (56 °C for 30 min), samples were serially diluted (fourfold) from a starting dilution of 1:4, and were incubated with equal volumes of 100 tissue culture infectious dose 50% (TCID50) of attack virus solution for 1 h at 37 °C, 5% CO_2_. Then, 0.1 mL 1.0~2.5 × 10^5^/mL cell suspension was added to each well, and incubated in a 37 °C, 5% CO_2_ incubator to culture; the results were judged at the 4th day. The neutralization endpoint was calculated using Karber’s method (serum dilution was converted to logarithm), i.e., the highest dilution of serum that could protect 50% of cells from 100 TCID50 challenge virus infection was the antibody titer of that serum.

### 2.7. Cytokine Analysis Assay

On days 46 and 71 after immunization, 10 rats (5 male rats and 5 female rats) were randomly selected from the negative control group, low-dose vaccine group or high-dose vaccine group. Serum was isolated from peripheral blood which collected in those groups. After treating with CBA (Cytometric Bead Array) kit (Biolegend, San Diego, CA, USA), the expression levels of IL-2, IL-6, TNF-α and IFN-γ in serum were analyzed by flow cytometry (BD LSRESSATM).

### 2.8. Vaccine Safety Evaluation

For vaccine safety evaluation, Sprague Dawley rats were randomly divided into five groups (15 male and 15 female rats in each group): physiological saline (as a negative control), low-dose (1 dose/rat) alum/CpG adjuvant, high-dose (3 doses/rat) alum/CpG adjuvant, low-dose (1 dose/rat) alum/CpG adjuvanted vaccine and high-dose (3 doses/rat) alum/CpG adjuvanted vaccine groups. Rats were immunized quartic at 2 week intervals.

During the whole experimental process, the mental states, behavioral activity, breathing, diet, water intake and local administration of the rats in each group were observed every day. In addition, the body weight and food intake of the rats in each group were detected every week. At the same time, body temperature was detected before the first immunization, 4–6 h after the first immunization (day 1), the next day after the first immunization (day 2), 4–6 h after the last immunization (day 43), the next day after the last immunization (day 44) and before the end of the recovery period (day 70). Three days after the last immunization (day 46) and after the recovery period (day 71), the body weight and organ weight of all rats were detected and recorded, and the organ weight/body weight ratio was calculated.

### 2.9. Pathological Examination

The organ tissues (heart, liver, spleen, lung, kidney and brain), sternal and femoral bone marrow, lymph nodes, groin and local administration of rats were removed at D46 (*n* = 10) and D71 (*n* = 5) after the quartic immunization (day 43) and fixed in 10% phosphate-buffered formalin for 24 h prior to paraffin embedding. Then, the paraffin sections (4–5 µm in thickness) were stained with hematoxylin–eosin. The sections were pictured and assessed by pathologists according to a 5-grade method (mild, mild, moderate, marked and severe).

### 2.10. Statistical Analysis

Statistical analyses were performed with two-tailed analysis, and the significance level was set at 0.05 or *p* ≤ 0.05. The data of antibody titers, cytokines, body weight, body temperature, food intake, organ/body weight ratios, etc., were statistically analyzed with SAS (9.2) and GraphPad prism (8.0) software, and the means and standard deviations (mean ± s.d) were calculated. The male and female animals were counted separately, and the differences between the test group and the control group were compared.

## 3. Results

### 3.1. Procedures for Immunization and Sample Collection

To evaluate the effect of CpG 7909 (abbreviated as CpG) in inactivated SARS-CoV-2 vaccine, we used a dual adjuvant system, including aluminum hydroxide (alum) and CpG in the inactivated SARS-CoV-2 vaccine (Figure 1A). As shown in Figure 1B, all rats were vaccinated four times on days 1, 15, 29, and 43. Blood samples from the experimental animals were collected at fixed time points and used to obtain antibody and cytokine data (Figure 1B).

### 3.2. Detection of Antibodies after Immunization in Rats

In order to test the immunogenicity of the alum/CpG-adjuvanted inactivated SARS-CoV-2 vaccine, we first tested the specific IgG antibodies in rats. Two weeks after the primary immunization, we detected specific IgG antibodies to the spike protein of SARS-CoV-2 in rat serum (Figure 2A). From the 15th to 71st days after primary immunization, specific IgG antibodies to the spike protein were detected in the two vaccine groups, and the levels of the specific IgG antibody titers increased gradually with the increases in administration time. At the end of the recovery period, the specific IgG antibody titers of the low-dose vaccine and high-dose vaccine groups were 25,600 and 102,400, respectively. The specific IgG antibody titers were not significantly different between male and female rats (*p* > 0.05).

Next, we tested neutralizing antibodies in SD rats (Figure 2B–D). Neutralizing antibodies in serum were detected at different detection time points, and the level of neutralizing antibodies gradually increased with the increase in the number of immunizations, reaching a peak at the final monitoring time point (day 71) (Figure 2C). The neutralizing antibody conversion rates of the vaccinated groups were all 100% (Figure 2D). At all monitoring time points after immunization, the levels of neutralizing antibodies produced by the vaccinated rats were significantly higher than those of the control group (Figure 2B).

### 3.3. Neutralizing Antibody Responses against SARS-CoV-2 Variants

To determine whether an inactivated alum/CpG-adjuvanted SARS-CoV-2 vaccine would cross-protect different SARS-CoV-2 variants, we measured the neutralizing antibodies induced by the vaccine against Beta, Delta and Omicron variants (Figure 3). At all the tested time points, high-dose and low-dose immunized rats produced neutralizing antibodies against the Beta variant (Figure 3A1). The neutralizing antibody level in the high-dose vaccine group was slightly higher than that in the low-dose vaccine group, but there was no significant difference (*p* > 0.05). Neutralizing antibody titers in the high-dose vaccine group reached the highest value at day 43, and the serum neutralizing antibody GMTs against the Beta variant reached an astonishing 6929 (Figure 3A2). The seroconversion rate of antibodies in all cases reached 100% (Figure 3A3). The serum neutralizing antibodies against the Delta and Omicron variants produced by the high-dose vaccine and low-dose vaccine groups were similar to those of the Beta variant (Figure 3B1–B3,C1–C3), but the corresponding neutralizing antibody titers were reduced. The neutralizing antibody titers of the high-dose vaccine group at day 43 had GMTs of 1732 and 795 for the serum neutralizing antibodies of the Delta and Omicron variants, respectively.

### 3.4. Cytokines

Compared with negative control, the levels of IL-2 and IL-6 at day 46 and levels of TNF-α and IFN-γ at day 71 after immunization were significantly higher in the low-dose vaccine group, and the levels of IL-2, IL-6, TNF-α and IFN-γ in peripheral blood were significantly increased at days 46 and 71 after immunization in the high-dose vaccine group. The levels of IL-6, TNF-α, and IFN-γ at day 46 and IL-2, IL-6, and TNF-α at day 71 after immunization in the high-dose vaccine group were significantly higher than those in the low-dose vaccine group, and the levels of the four cytokines reached the highest values at day 46 (Figure 4).

### 3.5. Safety

During the whole experiment, no mortality was observed in all the groups, no abnormal changes related to the adjuvant or vaccine were observed in clinical observation and no obvious abnormal changes were found in the administration sites of all rats. There was no difference in mean weekly body weight between all groups of rats during the experimental period (Appendix A). The body temperature of the rats in each group was in the normal range, and there was no significant change (Appendix A). The food intake of rats in each group was not affected by the type of immunization. The food intake of male rats was slightly higher than that of female rats, but there was no significant difference (Appendix A). Compared with the negative control group, the viscera-to-body ratios of the heart, liver, lung, kidney and brain of the rats in each group at day 46 did not show significant differences. The viscera-to-body ratios of the spleen of male and female rats in the high-dose vaccine group were similar; however, those of female rats in the low-dose adjuvant group were slightly increased (Figure 5A,B). There were no significant differences in the viscera-to-body ratios of the spleen of the rats in each group at day 71 (Figure 5C,D), which suggests that the change in the spleen weight could be basically recovered.

At day 46, 20 rats of each group (10 males and 10 females) were dissected; the remaining rats were dissected at day 71 for histopathological examination. There were no obvious abnormal changes visible to the naked eye in the administration sites of each rat, and the H&E staining of the heart, liver, lung, kidney, brain and other major organs of the rats in each group also showed no obvious abnormality (Figure 6). Compared with the negative control group, the number of granuloma cells was increased in the bone marrow (femur and sternum) in the other groups, and the germinal centers of the spleen white pulp in the vaccine group showed hyperplasia at day 46 (Appendix A). The results of H&E staining showed that the spleen white pulp in the high-dose vaccine group at day 46 had mild germinal center hyperplasia, and lesions of the bone marrow (femur and sternum) and spleen related to the vaccine or the adjuvant could be fully recovered at day 71 (Figure 6A). At day 46, mild cortical germinal center hyperplasia and medullary plasmacytosis were seen in high-dose adjuvant and high-dose vaccine rats (Appendix A), but the degree and incidence of inguinal lymph nodes were significantly reduced at day 71 (Figure 6B). At day 46, moderate granulomatous inflammation and mild interstitial edema were observed locally in the high-dose vaccine group and the high-dose adjuvant group. Statistical analysis showed that the lesion had a dose-dependent relationship (Appendix A); thus, it was considered to possibly be related to the adjuvant. However, at day 71, the degree of the lesion at the administration site was reduced, suggesting that there was a partial recovery (Figure 6C). In view of the mild lesions and recovery of the lesions in each group of rats, the above pathological changes are non-injurious, and are not toxic reactions caused by the adjuvant or vaccine.

## 4. Discussion

A variety of vaccines, such as mRNA vaccines, Adenovector based vaccines, and inactivated vaccines have been developed against SARS-CoV-2 and have shown efficacy [3,32]. Adenoviral vector vaccines and inactivated vaccines belong to the best studied and most utilised platforms. Although vaccines based on mRNA technology platforms are relatively new and mRNA vaccines against SARS-CoV-2 were the first authorized mRNA-based vaccines, mRNA vaccines generate higher levels of neutralizing antibody responses than inactivated or adenovector based vaccines [33]. Instability, high innate immunogenicity and delivery issues were the main obstacles of mRNA vaccines, and there are still many aspects of mRNA vaccines that need to be understood to assess the quality of the immune response elicited by the vaccine. mRNA from vaccinations can be found in germinal centers potentially disturbing the general adaptive immune responses [34]. Inactivated vaccines have a mature process and are safe, but compared with mRNA vaccines, inactivated vaccines have a longer production cycle and weaker immunogenicity. mRNA and adenovector based vaccinations are the uncontrollable concentrations of the spike protein expression leading potentially to toxicological problems [12], but for inactivated vaccines, it is possible to know exactly how much active substance concentration was injected.

Variants have been emerging since the beginning of the COVID-19 pandemic, including Alpha (B.1.1.7), Beta (B.1.351), Gamma (P.1), Delta (B. 1.617.2) and Omicron (B.1.1.529). The latest variant of concern, Omicron, is spreading rapidly around the world, with record reported morbidity. Mutations in the receptor binding domain (RBD) on the spike protein (S) directly lead to enhanced immune evasion ability of the virus, especially for the Omicron variant, which has a large number of mutation sites in the RBD, increasing the infectivity and pathogenic potential of the virus, and reducing the neutralizing ability of the vaccine to induce antibodies [11,35,36]. Therefore, developing a COVID-19 vaccine with high immunogenicity and safety against SARS-CoV-2 variants is essential to control the global COVID-19 pandemic and prevent further illness and fatalities. Here, we report a new adjuvanted inactivated SARS-CoV-2 vaccine candidate, which was formulated with aluminum hydroxide (alum) and CpG, that induces high levels of neutralizing antibody titers to provide protection against SARS-CoV-2 and shows a good safety in rats.

In our study, both novel low-dose and high-dose adjuvanted inactivated SARS-CoV-2 vaccine with CpG and Alum induced high levels of specific IgG antibodies and neutralizing antibodies against SARS-CoV-2, and the continuous stability of high-level neutralizing antibody titers over time shows that it has good, long-lasting immunity properties. Moreover, the vaccine exhibited equal effectiveness against the variants Beta, Delta and Omicron through two immunization doses in rats, which demonstrates its immune spectral property. Positive correlations have been shown between virus-specific IgG antibody titers and COVID-19 severity. The SARS-CoV-2-specific IgG antibody is a marker of COVID-19 infection or vaccination and helps to monitor and control the spread of COVID-19. In the evaluation of vaccine effectiveness, neutralizing antibody titers play an irreplaceable role as an important evaluation index [37,38]. They are all closely related to humoral immunity [39]. In addition to specific IgG antibodies and neutralizing antibodies, the vaccine promoted the upregulation of cytokines in serum. Although specific T cells were not analyzed, cytokine expression is associated with cellular immunity. However, the immune regulation system is an extremely complex network, and the mechanism of action of the vaccine to prevent new coronavirus morbidity needs more in-depth research [40].

Pre-clinical safety evaluations are one of the key phases in the development of new vaccines. To expand the safety analysis of our vaccine, we decided to examine its effects on rats in detail. The well-characterized rat model allows for a more detailed clinical and histopathological exploration of changes in different tissues. To provide sufficient evidence of toxicity that the proposed dose of alum/CpG-adjuvanted inactivated SARS-CoV-2 vaccine may have caused, as many indicators as possible were tested in repeated-dose toxicity studies. The results of this study showed that there were no significant changes in food intake, body weight and body temperature in rats. Eye examination and urinalysis revealed no abnormalities (data not shown). After repeated immunizations, no specific affected organs were found. Histopathological examination of immune organs showed lesions in the bone marrow, spleen and inguinal lymph nodes, but the related lesions disappeared during the recovery period. This further indicates that the vaccine has no obvious immunotoxicity for rats. Histopathological examination, including brain, liver, kidney, etc., showed no significant changes. Mild inflammation occurred in the local area of administration, but this pathological change is not invasive and is not a toxic reaction caused by the vaccine or adjuvant. Therefore, preclinical testing has shown that the prepared alum/CpG adjuvanted inactivated COVID-19 vaccine demonstrated a favorable safety profile.

In face of the severe challenge of SARS-CoV-2 mutations against existing antibodies and vaccines, the development of an alum/CpG adjuvanted inactivated COVID-19 vaccine provides a potential solution for the COVID-19 pandemic.

## Figures and Tables

**Figure 1 vaccines-10-01208-f001:**
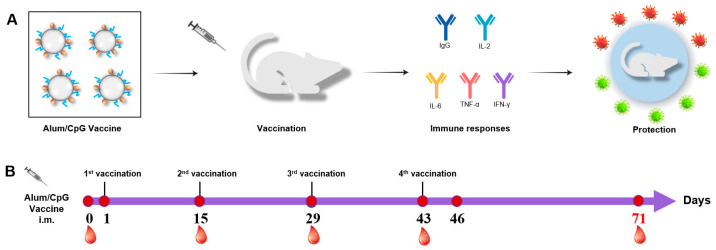
The immunization strategy foe SD rats. (**A**) Flow chart of preparation and (**B**) the immunization strategy for CpG-adjuvanted inactivated SARS-CoV-2 vaccine in rats.

**Figure 2 vaccines-10-01208-f002:**
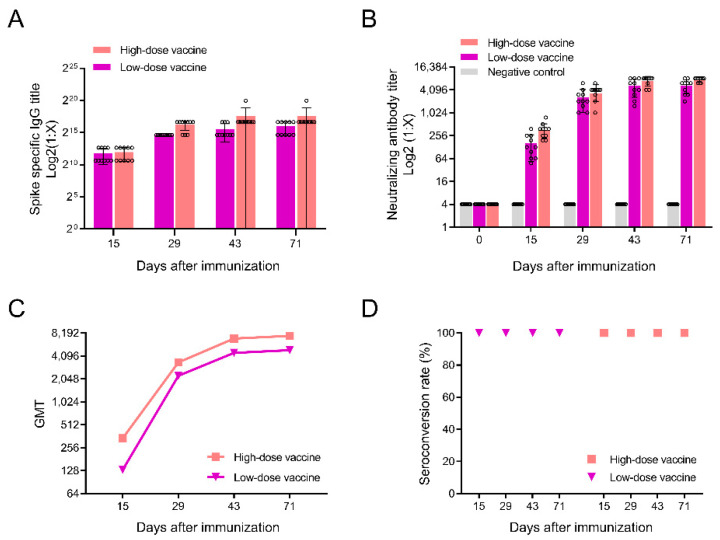
The antibody responses of immune serum to SARS-CoV-2. (**A**) Results of the specific IgG antibodies to SARS-CoV-2 spike protein in serum at different time points detected by ELISA, each circle represents a sample; (**B**) results of the levels of neutralizing antibodies against SARS-CoV-2 in serum, each circle represents a sample; (**C**) measurement of the GMTs of neutralizing antibody; (**D**) neutralizing antibody seroconversion rates in the serum of immunized rats were measured. The data are presented as means ± s.d (*n* = 10).

**Figure 3 vaccines-10-01208-f003:**
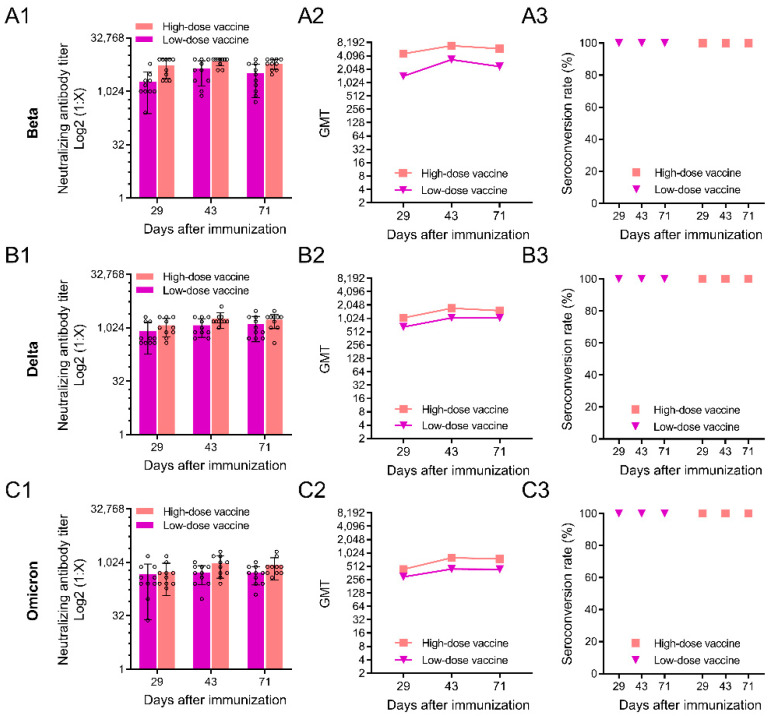
Results of neutralizing antibody responses against SARS-CoV-2 variants. (**A1**–**A3**), The neutralizing antibody levels against the Beta variant in serum and the GMTs of neutralizing antibodies and the seroconversion rates of neutralizing antibodies, respectively; (**B1**–**B3**) the neutralizing antibody levels against the Delta variant in serum and the GMTs of neutralizing antibodies and the seroconversion rates of neutralizing antibodies, respectively; (**C1**–**C3**) the neutralizing antibody levels against the Omicron variant in serum and the GMTs of neutralizing antibodies and the seroconversion rates of neutralizing antibodies, respectively. The data are presented as the means ± s.d (*n* =10).

**Figure 4 vaccines-10-01208-f004:**
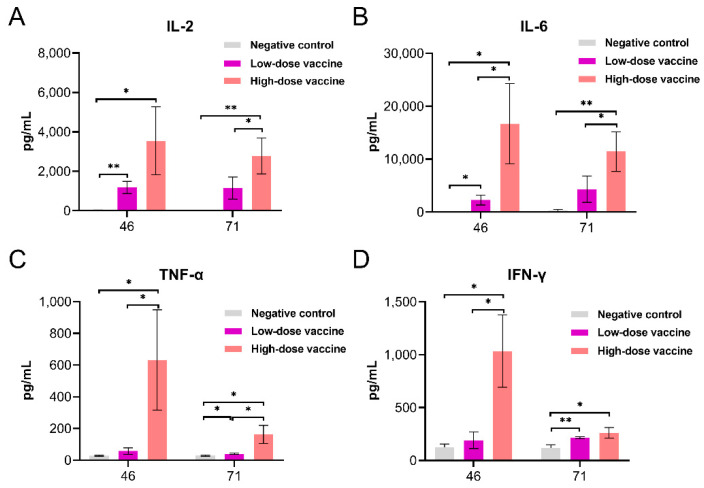
Expression of cytokines in peripheral blood after immunizations. (**A**) IL-2, (**B**) IL-6, (**C**) TNF-α and (**D**) IFN-γ. The data are presented as the means ± s.d. (*n* = 10), * *p* < 0.05, ** *p* < 0.01.

**Figure 5 vaccines-10-01208-f005:**
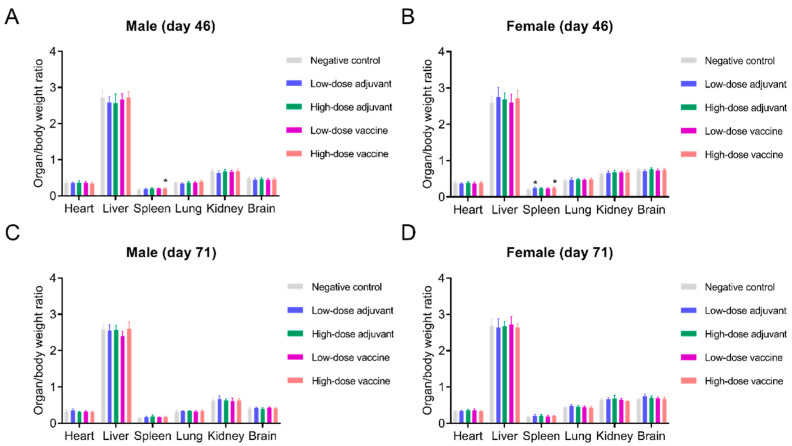
Clinical evaluation of rats in different groups. (**A**,**B**) Organ/body weight ratios at day 46 (*n* = 10). Organ/body weight ratios at day 46 (*n* = 10). (**C**,**D**) Organ/body weight ratios at day 71 (*n* = 5), * *p* < 0.05.

**Figure 6 vaccines-10-01208-f006:**
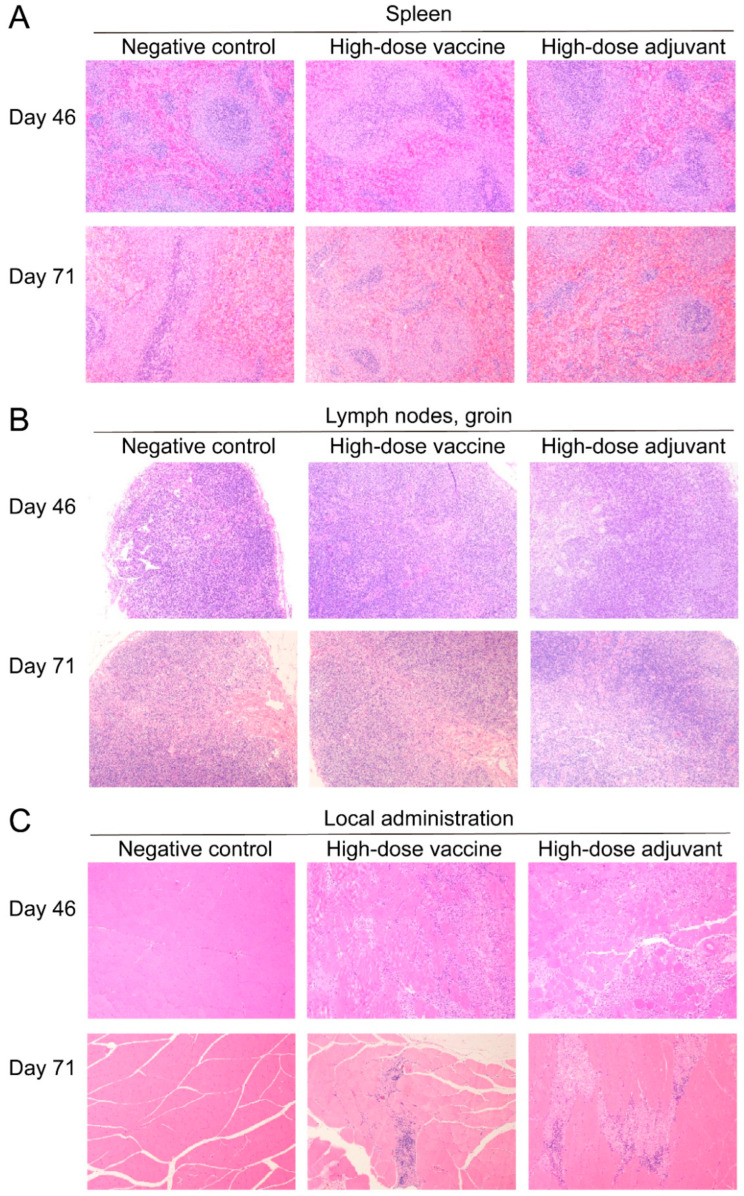
The results of the histopathological examinations for pathological changes in (**A**) spleen, (**B**) lymph nodes and groin and (**C**) local administration. H&E, 100×.

## Data Availability

Not applicable.

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
