# Peer review of "Alum/CpG Adjuvanted Inactivated COVID-19 Vaccine with Protective Efficacy against SARS-CoV-2 and Variants"

_vaccines, 2022, doi:10.3390/vaccines10081208_

Round 1

Reviewer 1 Report

An important factor for scientists is to discern what is supported by scientific facts and what is not: this is also why I think the authors need to rewrite their manuscript, and the major focus of correction should be on the introduction and the discussion.

The authors measure components like specific IgG-responses that do not have a real bearing in the initial infection and propagation of SARS-COV-2. It would have been more important to measure the IgA levels and how these pretty unspecific Ig are influenced by the vaccinations. Of course, I do not blame the authors as the exact same strategy has been applied in many recently published manuscripts to show effectiveness of the vaccinations. I like that the authors tested the neutralizing IgG antibody capacity against the different virus variants. But, what does this really mean in reality?

Here is why I think introduction and discussion need to be rewritten. 

It is commonly understood that a few aa exchanges in the spike protein will not majorly alter the recognition of a polyclonal immune response that has been previously established e.g. by vaccination. Most virologists and epidemiologists  claim that variants are important in disease development of respiratory infects based on the original observations of the importance of HIV variants in AIDS. However, AIDS  is not comparable to respiratory infect caused diseases.

It is also clear from the human infection trial with the predominant SARS-COV-2 variant at the beginning of the pandemic (https://www.nature.com/articles/s41591-022-01780-9) that only the upper respiratory tract is targeted by SARS-COV-2 infections in healthy individuals. The responsible research scientist was asked what he would expect if the healthy individuals would have been infected with the more pathogenic Delta- variant of the virus. He replied nothing different from what he also saw with the other virus variant.

There are other aspects about the pathogenicity and lethality of SARS-COV-2 mentioned in the discussion by the authors. Namely, the fatality rate number has to be taken with a grain of salt: firstly, most countries did not make a difference between with or on SARS-COV-2 diseased; secondly, the mass-testing by RT-PCR or viral antigen presence only indicated that the individual carried SARS-COV-2 in the mucus. These assays cannot show the infection of an individual with the virus and including whether  the disease was caused by another respiratory infect on top of the SARS-COV-2 present. The number of 6Mio. SARS-COV-2 deaths  itself  has to be put in correlation with the Spanish flu that was caused with high probability by a normal H1N1 virus. In 1918/19, this virus caused 25-80 times more deaths than the so-called “dangerous” SARS-COV-2.

In general, there are no randomized double blinded studies with vaccinated individuals in challenge studies available showing successful vaccination against a respiratory tract infect, e.g., for over 60 years it has not been possible to develop a much needed vaccine that is effective against RSV, another respiratory viral infect. Yes, in many studies, research scientists showed that they indeed were able after vaccination to identify specific IgG immune responses or even specific T-cell activation and memory T-cell responses. However, what about the IgA immune responses of the innate immune response?

It is also clear that the major defense against respiratory tract infection is not dependent on the adaptive immune responses. It is clearly dependent on the MALT and BALT immune responses of unspecific IgG and IgA and atypical T-cell responses and other innate immune responses. Only in later stages of disease, when the disease becomes acute/severe it may or may not be desirable to be vaccinated. I know of one recent Cell publication (https://www.ncbi.nlm.nih.gov/pmc/articles/PMC8786601/) that shows that the IgA levels are hardly affected by the mRNA COVID-19 vaccines. As a major importance for respiratory infection is the balance between tolerance and immune response. It could well be that an activated adaptive immune response is not desirable, which may lead to a more immediate aggressive immune response furthering the destruction of the respiratory tract.

In the Introduction the authors make a statement with no factual evidence:

In the line 55 the authors state: “Until now, COVID-19 vaccines have been proved to be effective and show strong safety profiles. A sentence without real evidence from randomized double blinded studies that it is indeed effective preventing infections and that this new vaccinations strategies  have strong safety profile. With this sentence the authors make a real disservice to their publication. 

Additionally, this statement is the opposite from what is so often heard when one analyzes the latest scientific publications. For the effectivity ,see above explanations and for the safety profiles: the authors should consider this - what is safe in a drug when one does not know the end-concentrations of the active substance? The expressed dose varies from person to person and the whole vaccination becomes a toxicological nightmare. E.g., the active substance, the spike protein, varies between 0-174pg/ml plasma in vaccinated individuals, in severely sick individuals, it is about 74pg/ml plasma and, no consideration has been given to individual spike protein pools (https://www.ncbi.nlm.nih.gov/pmc/articles/PMC8786601/). What about the disturbance of GCs in immune suppressed individuals by mRNA vaccinations (https://www.ncbi.nlm.nih.gov/pmc/articles/PMC8808747/)? Is this because these immune suppressed individuals have an already weakened immune system that it is further destabilized by the mRNA from vaccination as the mRNA is found in the GC for at least 6 weeks (https://www.ncbi.nlm.nih.gov/pmc/articles/PMC8786601/) and polynucleosides/tides (pGC) can cause inflammations?

A comparison of the authors vaccine strategy to that of the mRNA and adenovector based vaccines is urgently needed.

By comparison of different existing databases and studies one can find that severe vaccination side effects of the new vaccinations can amount up to 0.8% of vaccinated people. If one puts this in relation to the swine flu vaccination program that was pulled from the market in 1976 on the basis of 362 cases of severe side-effects correlated with the vaccination of 45 Mio people, this would account for about 0.001% severe side effects in correlation with vaccination. This would be 800 less severe side effects as seen today with the new vaccination strategies.

Please see also minor criticism:

Under point 2.5 within M & M you would need to indicate GMT (geometric mean titer).

In general what happened to the adjuvants control alone?  Please, indicate them in the figures of Figure 2.

In Figure 2B of the results did you mix the groups of adjuvants only (low or high concentrations) with the indicated negative control and stated this in the figure as negative control?

I assume in Figure 2D the low and high dose vaccinations overlap? Please find a way how this can be better distinguished.

In Figure 7, day 71 in the local administration tissue: are there not indications of inflammatory responses (invasion of lymphocytes) in the high-d vaccine and high-d adjuvant?

Author Response

Reviewer 1

Comments to the Author

The manuscript entitled “Alum/CpG adjuvanted inactivated COVID-19 vaccine with protective efficacy against SARS-CoV-2 and variants” and authored by Zhang et al., set out to developed a COVID-19 vaccine, base on inactivated SARS-CoV-2 virus and an adjuvant consisting of aluminum hydroxide (alum) and CpG. The authors tested the immunogenicity and safety of the developed vaccine in Sprague Dawley rats, and demonstrated that the vaccine elicited high titers of SARS-CoV-2 spike-specific IgG antibody and neutralizing antibody. The antibodies were also shown to have cross-neutralization activities against the Beta, Delta and Omicron SARS-CoV-2 variants, in addition to the original SARS-CoV-2 strain. The manuscript has some weaknesses and will require a major revision before consideration for publication.

 Major Comments:

  1. The authors measure components like specific IgG-responses that do not have a real bearing in the initial infection and propagation of SARS-COV-2. It would have been more important to measure the IgA levels and how these pretty unspecific Ig are influenced by the vaccinations. Of course, I do not blame the authors as the exact same strategy has been applied in many recently published manuscripts to show effectiveness of the vaccinations. I like that the authors tested the neutralizing IgG antibody capacity against the different virus variants. But, what does this really mean in reality?

Here is why I think introduction and discussion need to be rewritten. 

Response: We partly agree with the reviewer’s standpoint. However, in the previous paper by Yan et al. (2022) and Moradi et al. (2021) showed the positive correlation between virus‐specific IgG antibody titers and COVID‐19 severity. Thus, the SARS-CoV-2-specific IgG antibody is an excellent marker of COVID-19 infection or vaccination and helps monitor and control COVID-19 spread. Variants have been emerging since the beginning of COVID-19 pandemic, including Alpha (B.1.1.7), Beta (B.1.351), Gamma (P.1), Delta (B. 1.617.2), Omicron (B.1.1.529). The latest variant of concern, Omicron, is spreading swiftly around the world with record morbidity reports. However, in the previous paper by Yaniv et al. (2022) showed a cryptic circulation of the Delta variant even with the increased levels of Omicron variant. Interestingly, Yaniv et al predict that if the Omicron levels decrease until eliminated, the mentioned cryptic circulation may result in the reemergence of a Delta morbidity wave or in the possible generation of a new threatening variant. In order to evaluate the effectiveness of vaccines against COVID-19 variants, we tested the neutralizing IgG antibody capacity against the different virus variants.

  1. An important factor for scientists is to discern what is supported by scientific facts and what is not: this is also why I think the authors need to rewrite their manuscript, and the major focus of correction should be on the introduction and the discussion.

It is commonly understood that a few aa exchanges in the spike protein will not majorly alter the recognition of a polyclonal immune response that has been previously established e.g. by vaccination. Most virologists and epidemiologists  claim that variants are important in disease development of respiratory infects based on the original observations of the importance of HIV variants in AIDS. However, AIDS is not comparable to respiratory infect caused diseases.

It is also clear from the human infection trial with the predominant SARS-COV-2 variant at the beginning of the pandemic (https://www.nature.com/articles/s41591-022-01780-9) that only the upper respiratory tract is targeted by SARS-COV-2 infections in healthy individuals. The responsible research scientist was asked what he would expect if the healthy individuals would have been infected with the more pathogenic Delta- variant of the virus. He replied nothing different from what he also saw with the other virus variant.

There are other aspects about the pathogenicity and lethality of SARS-COV-2 mentioned in the discussion by the authors. Namely, the fatality rate number has to be taken with a grain of salt: firstly, most countries did not make a difference between with or on SARS-COV-2 diseased; secondly, the mass-testing by RT-PCR or viral antigen presence only indicated that the individual carried SARS-COV-2 in the mucus. These assays cannot show the infection of an individual with the virus and including whether  the disease was caused by another respiratory infect on top of the SARS-COV-2 present. The number of 6Mio. SARS-COV-2 deaths  itself  has to be put in correlation with the Spanish flu that was caused with high probability by a normal H1N1 virus. In 1918/19, this virus caused 25-80 times more deaths than the so-called “dangerous” SARS-COV-2.

In general, there are no randomized double blinded studies with vaccinated individuals in challenge studies available showing successful vaccination against a respiratory tract infect, e.g., for over 60 years it has not been possible to develop a much needed vaccine that is effective against RSV, another respiratory viral infect. Yes, in many studies, research scientists showed that they indeed were able after vaccination to identify specific IgG immune responses or even specific T-cell activation and memory T-cell responses. However, what about the IgA immune responses of the innate immune response?

It is also clear that the major defense against respiratory tract infection is not dependent on the adaptive immune responses. It is clearly dependent on the MALT and BALT immune responses of unspecific IgG and IgA and atypical T-cell responses and other innate immune responses. Only in later stages of disease, when the disease becomes acute/severe it may or may not be desirable to be vaccinated. I know of one recent Cell publication (https://www.ncbi.nlm.nih.gov/pmc/articles/PMC8786601/) that shows that the IgA levels are hardly affected by the mRNA COVID-19 vaccines. As a major importance for respiratory infection is the balance between tolerance and immune response. It could well be that an activated adaptive immune response is not desirable, which may lead to a more immediate aggressive immune response furthering the destruction of the respiratory tract.

In the Introduction the authors make a statement with no factual evidence:

In the line 55 the authors state: “Until now, COVID-19 vaccines have been proved to be effective and show strong safety profiles“. A sentence without real evidence from randomized double blinded studies that it is indeed effective preventing infections and that this new vaccinations strategies  have strong safety profile. With this sentence the authors make a real disservice to their publication. 

Additionally, this statement is the opposite from what is so often heard when one analyzes the latest scientific publications. For the effectivity ,see above explanations and for the safety profiles: the authors should consider this - what is safe in a drug when one does not know the end-concentrations of the active substance? The expressed dose varies from person to person and the whole vaccination becomes a toxicological nightmare. E.g., the active substance, the spike protein, varies between 0-174pg/ml plasma in vaccinated individuals, in severely sick individuals, it is about 74pg/ml plasma and, no consideration has been given to individual spike protein pools (https://www.ncbi.nlm.nih.gov/pmc/articles/PMC8786601/). What about the disturbance of GCs in immune suppressed individuals by mRNA vaccinations (https://www.ncbi.nlm.nih.gov/pmc/articles/PMC8808747/)? Is this because these immune suppressed individuals have an already weakened immune system that it is further destabilized by the mRNA from vaccination as the mRNA is found in the GC for at least 6 weeks (https://www.ncbi.nlm.nih.gov/pmc/articles/PMC8786601/) and polynucleosides/tides (pGC) can cause inflammations?

A comparison of the authors vaccine strategy to that of the mRNA and adenovector based vaccines is urgently needed.

By comparison of different existing databases and studies one can find that severe vaccination side effects of the new vaccinations can amount up to 0.8% of vaccinated people. If one puts this in relation to the swine flu vaccination program that was pulled from the market in 1976 on the basis of 362 cases of severe side-effects correlated with the vaccination of 45 Mio people, this would account for about 0.001% severe side effects in correlation with vaccination. This would be 800 less severe side effects as seen today with the new vaccination strategies.

Response: Thank you for the comments. It is our great appreciation to your kindly dealing with our “Alum/CpG adjuvanted inactivated COVID-19 vaccine with protective efficacy against SARS-CoV-2 and variants”. We have given some explanations for these comments and the manuscript has also been revised according to your valuable suggestions.

We thought that neutralizing antibodies elicited by prior infection or vaccination are likely to be key for future protection of individuals and populations against SARS-CoV-2 (https://www.ncbi.nlm.nih.gov/pmc/articles/PMC7723407/, https://www.ncbi.nlm.nih.gov/pmc/articles/PMC7836116/) and global herd immunity will help end the COVID-19 (pandemichttps://www.ncbi.nlm.nih.gov/pmc/articles/PMC8537265/). But it seems that the correlates of immunological protection from SARS-CoV-2 infection following vaccination or prior infection are still under investigation (https://www.ncbi.nlm.nih.gov/pmc/articles/PMC8786601/). Therefore, we make a more conservative statement based on the factual evidence in the introduction part. Almost all types of vaccines have been reported to be associated with adverse events, such as fever, headache, coughing, loss of appetite, vomiting, diarrhea, joint pain, and autoimmune conditions, most of which are mild and serious conditions are rare (https://www.ncbi.nlm.nih.gov/pmc/articles/PMC8381833/, https://www.ncbi.nlm.nih.gov/pmc/articles/PMC8315897/). It is difficult to see how mid- and long-term safety testing for the proposed vaccine can be performed in such a compressed time frame, but it has been proved that COVID-19 vaccines have an acceptable short-term safety profile (https://www.ncbi.nlm.nih.gov/pmc/articles/PMC8315897/). Facing the novel coronavirus SARS-CoV-2, the benefits of COVID-19 vaccination outweigh the risks, despite rare serious adverse effects (https://www.ncbi.nlm.nih.gov/pmc/articles/PMC8548286/). That’s why the COVID-19 vaccine strategies have been deployed globally. And we agree with you that “Until now, COVID-19 vaccines have been proved to be effective and show strong safety profiles” is a sentence without real evidence from randomized double blinded studies. We have rewrite the sentence in the introduction. Since the outbreak of COVID-19, several response strategies, including accelerating massive rollouts of current vaccines, increasing vaccine immunogenicity by increasing vaccination doses, and accelerating next-generation vaccines against variants, have been suggested. The existing vaccines such as mRNA vaccines, adenovector based vaccines and inactivated vaccines had been proved have the efficacy against SARS-CoV-2, but fewer uncommon although serious adverse events are observed after homologous or heterologous vaccination immunized with inactivated vaccines (https://pubmed.ncbi.nlm.nih.gov/34696271/). Therefore, we have modified existing inactive vaccines to increase their resistance to original strain as well as mutants.

we have rewritten the Introduction and Discussion section and the corresponding content has been revised in the manuscript. We have responded to related questions in question 1 and question.

Please see also minor criticism:

1.Under point 2.5 within M & M you would need to indicate GMT (geometric mean titer).

Response: Thank you for the comment. We have indicated GMT under point 2.5 within M&M.

2.In general what happened to the adjuvants control alone?  Please, indicate them in the figures of Figure 2.

Response: Thank you for the comment. In general, adjuvants do not cause the body to produce an immune response to a specific antigen, and accordingly, the body does not produce specific antibodies and neutralizing antibodies. So we did not set up a adjuvant control alone group.

3.In Figure 2B of the results did you mix the groups of adjuvants only (low or high concentrations) with the indicated negative control and stated this in the figure as negative control?

Response: Thank you for the comment. we did not set up adjuvant control alone (low or high concentrations) group in Figure 2B, and the group animals were immunized with physiological saline as negative control.

4.I assume in Figure 2D the low and high dose vaccinations overlap? Please find a way how this can be better distinguished.

Response: Thank you for the comment. The neutralizing antibody seroconversion rate in the two groups were all 100% at different detection time points, so they overlapped. In order to show them more clearly, we have modified the figure 2D.

Revised figure 2:

5.In Figure 7, day 71 in the local administration tissue: are there not indications of inflammatory responses (invasion of lymphocytes) in the high-d vaccine and high-d adjuvant?

Response: Thank you for the comment. Mild inflammation will occur in the local area of ad-ministration, but this pathological change is not invasive and is not a toxic reaction caused by the vaccine or adjuvant.

Reviewer 2 Report

The manuscript entitled “Alum/CpG adjuvanted inactivated COVID-19 vaccine with protective efficacy against SARS-CoV-2 and variants” and authored by Zhang et al., set out to developed a COVID-19 vaccine, base on inactivated SARS-CoV-2 virus and an adjuvant consisting of aluminum hydroxide (alum) and CpG. The authors tested the immunogenicity and safety of the developed vaccine in Sprague Dawley rats, and demonstrated that the vaccine elicited high titers of SARS-CoV-2 spike-specific IgG antibody and neutralizing antibody. The antibodies were also shown to have cross-neutralization activities against the Beta, Delta and Omicron SARS-CoV-2 variants, in addition to the original SARS-CoV-2 strain. The manuscript has some weaknesses and will require a major revision before consideration for publication.

 Major Comments:

1.      The authors did not mention how the inactivated SARS-CoV-2 virus and/or the adjuvants independently or together as a vaccine was quantified. It was therefore not clear what a dose of vaccine is, as has been used throughout the manuscript.

2.      It is not clear how many rats were used for the different sections of the study (Vaccine Immunogenicity Analysis and Vaccine Safety Analysis), and how the animals were grouped.

Section 2.4. suggests that there were three groups of 10 rats (5 males and 5 females) each that were injected with the vaccine at two-week intervals, but Line 106 says “The group animals were immunized with Physiological saline (as negative control), low dose vaccine (1 dose/rat) or high dose vaccine (3 dose/rat), respectively.” If all the groups were given the “test vaccine” how/where does the control group come in? It is confusing.

Section 2.8 suggests that were five groups of 30 rats (15 males and 15 females) each were injected with physiological saline, low dose, and high dose. How did two different doses and a control go into five groups of animals?

3.     Under 3.2. (Detection of antibodies after immunization) of the Results section, the authors said “The antibody titers in the high-dose vaccine group were slightly higher than those in the low-dose vaccine at each time point”, but the specific IgG antibody was four time more in the high dose group than in the low dose group. The authors would have to provide the actual titres for the different periods as Supplementary data.

4.     The authors should find a way to re-number Fig. 3; say, A(1), A(2), and A(3) for Beta, so that Beta is A, Delta is B, and Omicron is C.

I noticed that for the entire Fig. 3, the authors decided to exclude Day 15 after immunization. Why was that so?

5.  Under 3.5 (Safety) of the Results section, there was no significant difference in all the parameters that were observed and presented in Fig. 5. I will suggest that Fig. 5 is put in the Supplementary Information.

On Line 248, the authors made reference to “other major organs”. However, the only organ left apart from the ones that were mentioned (heart, liver, lung, kidney, and brain) is spleen. So, why did the authors say “other major organs”?

6.      Fig. 7 is not clear enough to see the pathological observations that were being described in the Results text for that figure, and the title of the figure would have to be revised.

In the legend of the same Fig. 7, I see the sentence/comment “This section may be divided by subheadings. It should provide a concise and precise description of the experimental results, their interpretation, as well as the experimental conclusions that can be drawn.” That I do not know how that got there.

7.      The two main paragraphs of the Discussion section have repetitions of the Results. In my opinion, the results were not appreciably discussed.

8.      I inferred from the Results that the three Supplementary Tables were for 46-day periods, but there was indication of this whatsoever in the Supplementary Tables.

Minor Comments:

1.      The use of the definite article “the” should be looked at; the article is used when something has already been mentioned, or if that thing is common/mundane.

2.      The use of abbreviations like VOC and CpG in the Abstract section should be defined first. The acronym “GMT” was never defined in the entire manuscript.

3.   The ethical approval number from IACUC should be provided under 2.1. (Ethics Statement) of the Materials and Methods section.

4.      There should be spaces between words that come before and after brackets, and there should also be spaces between numbers and their units. The only unit that is not spaced with numbers is the %.

5.      There are sentences in the manuscript that are not clear. Examples are the sentences in Lines 117-119 and 127-130.

6.      The use of the word “respectively” is not correct in most cases, as well as the use of the word “abbreviat” in Lines 99 and 170. These should be checked.

7. There were significant typographical and grammatical errors in the manuscript. I will suggest that the manuscript is reviewed by a native English-speaking person.

Author Response

Reviewer 2

Comments to the Author 

The manuscript entitled “Alum/CpG adjuvanted inactivated COVID-19 vaccine with protective efficacy against SARS-CoV-2 and variants” and authored by Zhang et al., set out to developed a COVID-19 vaccine, base on inactivated SARS-CoV-2 virus and an adjuvant consisting of aluminum hydroxide (alum) and CpG. The authors tested the immunogenicity and safety of the developed vaccine in Sprague Dawley rats, and demonstrated that the vaccine elicited high titers of SARS-CoV-2 spike-specific IgG antibody and neutralizing antibody. The antibodies were also shown to have cross-neutralization activities against the Beta, Delta and Omicron SARS-CoV-2 variants, in addition to the original SARS-CoV-2 strain. The manuscript has some weaknesses and will require a major revision before consideration for publication.

 Major Comments:

  1. The authors did not mention how the inactivated SARS-CoV-2 virus and/or the adjuvants independently or together as a vaccine was quantified. It was therefore not clear what a dose of vaccine is, as has been used throughout the manuscript.

Response: We greatly appreciate the reviewer’s comments. As per the reviewer’s suggestion, the method was provided in detail in the revised manuscript (Section 2.3 on Page 3): “The final concentrations of aluminum hydroxide and CpG were 0.45 mg/mL and 20 μg/mL, respectively. The inactivated SARS-CoV-2 virus concentration was 13 U/mL. One dose was 0.5mL.

  1. It is not clear how many rats were used for the different sections of the study (Vaccine Immunogenicity Analysis and Vaccine Safety Analysis), and how the animals were grouped.

Response: On the basis of the reviewer’s suggestion, we made a table to make the animal groupings in different sections of the study clearer.

Vaccine Safety Analysis

Group

Dose/Rat

Number of

animals/Gender

The negative control group

0

15

The low-dose alum/CpG adjuvant group

1

15

The high-dose alum/CpG adjuvant group

3

15

The low-dose alum/CpG adjuvanted vaccine group

1

15

The high-dose alum/CpG adjuvanted vaccine group

3

15

Vaccine Immunogenicity Analysis

The negative control group

0

5

The low-dose alum/CpG adjuvanted vaccine group

1

5

The high-dose alum/CpG adjuvanted vaccine group

3

5

  1. Section 2.4. suggests that there were three groups of 10 rats (5 males and 5 females) each that were injected with the vaccine at two-week intervals, but Line 106 says “The group animals were immunized with Physiological saline (as negative control), low dose vaccine (1 dose/rat) or high dose vaccine (3 dose/rat), respectively.” If all the groups were given the “test vaccine” how/where does the control group come in? It is confusing.

Response: We thank the reviewer for pointing out our mistake “SD rats were randomly divided into three groups (5 male and 5 female rats in each group), which received four intramuscular injections of the test vaccine at two-week intervals”. We have made the revision as suggested by the reviewer.

  1. Section 2.8 suggests that were five groups of 30 rats (15 males and 15 females) each were injected with physiological saline, low dose, and high dose. How did two different doses and a control go into five groups of animals?

Response: I think the reviewer misunderstood our description in section 2.8. The two different doses include alum/CpG adjuvant t and alum/CpG adjuvanted vaccine, respectively. The Line 139 says “For vaccine safety evaluation, Sprague Dawley rats were randomly divided into five groups (15 male and 15 female rats in each group): Physiological saline (as negative control), low-dose (1 dose/rat) alum/CpG adjuvant, high-dose (3 dose/rat) al-um/CpG adjuvant, low-dose (1 dose/rat) alum/CpG adjuvanted vaccine and high-dose (3 dose/rat) alum/CpG adjuvanted vaccine groups.”

  1. Under 3.2. (Detection of antibodies after immunization) of the Results section, the authors said “The antibody titers in the high-dose vaccine group were slightly higher than those in the low-dose vaccine at each time point”, but the specific IgG antibody was four time more in the high dose group than in the low dose group. The authors would have to provide the actual titers for the different periods as Supplementary data.

Response: We agree with the reviewer. The sentence for “The antibody titers in the high-dose vaccine group were slightly higher than those in the low-dose vaccine at each time point” has been deleted in the revised manuscript.

  1. The authors should find a way to re-number Fig. 3; say, A(1), A(2), and A(3) for Beta, so that Beta is A, Delta is B, and Omicron is C.I noticed that for the entire Fig. 3, the authors decided to exclude Day 15 after immunization. Why was that so?

Response: Thank you for the comment. We re-numbered the Figure. 3, Beta is A, Delta is B, and Omicron is C. In Figure. 3A/3B/3C, the left panel presents the neutralizing antibody titer, the middle panel presents GMT (geometric mean titer), and the right panel presents the neutralizing antibody seroconversion rate.

We first tested antibody titers against SARS-CoV-2. As shown in Figure 2, when we detected neutralizing antibody, we found that the neutralizing antibody titer at Day 15 was relatively low, and there was no statistical difference between the high-dose group and the low-dose group, which implied that we might be able to exclude Day 15, during subsequently detected neutralizing antibody responses against SARS-CoV-2 variants.

Revised Figure.3:

Figure 3. Results of neutralizing antibody responses against SARS-CoV-2 variants. (A1-A3), The neutralizing antibody levels against the Beta variant in serum and the GMTs of neutralizing an-tibodies and the seroconversion rates of neutralizing antibodies, respectively; (B1–B3) the neutralizing antibody levels against the Delta variant in serum and the GMTs of neutralizing anti-bodies and the seroconversion rates of neutralizing antibodies, respectively; (C1-C3) the neutralizing antibody levels against the Omicron variant in serum and the GMTs of neutralizing an-tibodies and the seroconversion rates of neutralizing antibodies, respectively. The data are presented as the means ± s.d. (n =10).

  1. Under 3.5 (Safety) of the Results section, there was no significant difference in all the parameters that were observed and presented in Figure. 5. I will suggest that Figure. 5 is put in the Supplementary Information. On Line 248, the authors made reference to “other major organs”. However, the only organ left apart from the ones that were mentioned (heart, liver, lung, kidney, and brain) is spleen. So, why did the authors say “other major organs”?

Response: As per the reviewer’s comments, we have moved the Figure 5 to the Supplementary information as Supplementary Figure S1, referred to as Figure S1. The corresponding revision have been made in the revised manuscript. In addition, descriptions of other major organs have been removed after consideration.

  1. Fig. 7 is not clear enough to see the pathological observations that were being described in the Results text for that figure, and the title of the figure would have to be revised.In the legend of the same Fig. 7, I see the sentence/comment “This section may be divided by subheadings. It should provide a concise and precise description of the experimental results, their interpretation, as well as the experimental conclusions that can be drawn.” That I do not know how that got there.

Response: Thank you for the comment. We deleted the sentence “This section may be divided by subheadings…...”. We changed the title of the Figure to “The result of the histopathological examination for pathological changes”.

Revised Figure.7

Figure 7. The result of the histopathological examination for pathological changes in (A) spleen, (B) lymph nodes and groin and (C) local administration. H&E, 100 x.

  1. The two main paragraphs of the Discussion section have repetitions of the Results. In my opinion, the results were not appreciably discussed.

Response: Thank you for the remarks, we have rewritten the Discussion section and the corresponding content has been revised in the manuscript.

  1. I inferred from the Results that the three Supplementary Tables were for 46-day periods, but there was indication of this whatsoever in the Supplementary Tables.

Response: As per the reviewer’s comments, to ensure the clarity of the results in the supplementary information, we have added a period of 46 days for pathological observation on the three tables of the supplementary information.

Minor Comments:

  1. The use of the definite article “the” should be looked at; the article is used when something has already been mentioned, or if that thing is common/mundane.

Response: As per the reviewer’s comments, we have checked and revised the “the” in the manuscript using the track changes function of Microsoft Word.

  1. The use of abbreviations like VOC and CpG in the Abstract section should be defined first. The acronym “GMT” was never defined in the entire manuscript.

Response: Thanks for your comments, we have checked and defined the first occurrence of abbreviations in the text, VOC and GMT have been revised using the track changes function of Microsoft Word.

  1. The ethical approval number from IACUC should be provided under 2.1. (Ethics Statement) of the Materials and Methods section.

Response: Thanks for your comments, the ethical approval number from IACUC have provided in Section 2.1 as follows:“ACU21-2409”.

  1. There should be spaces between words that come before and after brackets, and there should also be spaces between numbers and their units. The only unit that is not spaced with numbers is the %.

Response: Thank you for the remarks. We have corrected the space mistakes in the manuscript, accordingly. Please note that the changes in the revised manuscript using the track changes function of Microsoft Word.

  1. There are sentences in the manuscript that are not clear. Examples are the sentences in Lines 117-119 and 127-130.

Response: Thank you for the remarks. We have revised these sentences in the manuscript, accordingly. Please note that the changes in the revised manuscript using the track changes function of Microsoft Word.

  1. The use of the word “respectively” is not correct in most cases, as well as the use of the word “abbreviat” in Lines 99 and 170. These should be checked.

Response: Thank you for the remarks. We have revised these words in the manuscript, accordingly. Please note that the changes in the revised manuscript using the track changes function of Microsoft Word.

  1. There were significant typographical and grammatical errors in the manuscript. I will suggest that the manuscript is reviewed by a native English-speaking person.

Response: Thank you for the remarks. We have revised the manuscript by a native English-speaking person. Please note that the changes in the revised manuscript using the track changes function of Microsoft Word.

Round 2

Reviewer 1 Report

To reiterate from my previous review: An important factor for scientists is to discern what is or is not supported by scientific evidences. The response to my initial concerns suggest the authors’ data presentation suffers from wide-ranging objectivity and healthy criticism and, therefore in my opinion, makes the manuscript immature for publication. Although  I noticed a certain improvement in general, many of my concerns still remain. I´ll try to clarify below with examples what I mean about objectivity. Using a convenient narrative should not be the criteria to easily publish. This is just damaging to science over-all and also to human health.

The authors did not address different points that I considered essential in the rebuttal nor in the corrected manuscript, e.g.,: what about the lack of interaction of the adaptive immune system with initial infection and propagation of SARS-COV-2? In other words, what is the role of the adaptive immune system that we can specifically activate through vaccination in respiratory tract infections? Instead, the authors cited publications that relied on mined datasets built mainly on RT-PCR data. I know of at least three well developed western countries that do not have the numbers to distinguish who were admitted because of COVID19 from the hospitalization cases. It is naïve to think that we have any reliable data worldwide to calculate statistically whether these new vaccinations have any positive effect. And the companies responsible for development of these vaccines did not do the randomized double blind studies with challenges that would be required to obtain answers about the absolute effectivity of these vaccinations. Also, it is clear from the absence of any effective vaccination against RSV, another viral respiratory tract infect, that development of vaccinations against respiratory tract infects have not been successful for over 60 years.

Science needs to be depoliticized and as such I do not think that the WHO numbers should be used in any scientific publication since e.g. starting with confusion about the ambiguity of the case definition that led up to the numbers. Panic driven governments decided to use the SARS-COV-2 RT-PCR assay as the standard neglecting the possibility of adenovirus or  bacterial respiratory tract infections co-infections, which are frequently observed together with coronavirus infections. Here, I would remind the authors about the mild pathogenicity of SARS-COV-2 in the human infection trials (Killingley et al., 2022). Additionally, the RT-PCR method on its own will only show a particular part of the virus and will not be able to define its infectivity. Furthermore, the RT-PCR assay or the antigen assays in use can only find virus that is trapped in the mucus of the innate immune system.

Even more basic should be a ROC analysis, which is missing worldwide: a statistical view on sensitivity and specificity of the used assays in the pandemic. This analysis was not done in the pandemic management to answer the question of how good the RT-PCR can detect an infected person in a human population. This is a major concern if e.g. one tests 10´000 people and let´s say the RT-PCR assay were fantastic with a high sensitivity of 99% and a 100% specificity, one would still find that 100 people among the 10´000 would be indicated as false RT-PCR positives. I doubt very much from available evidences that any of the used assays come even close to this sensitivity and specificity. These numbers therefore are useless and misleading.

Where I do see the usefulness of the data in present manuscript is the data using a full inactivated SARS-COV-2 variant with alum and poly-CpG in vaccination to produce serum to neutralize all major SARS-COV-2 variants. However, there is a lack of objectivity on the part of the authors because even with the description of the findings of other research groups it seems that the humoral response is the only important adaptive immune response for the authors of present manuscript. This of course we know is wrong as the adaptive cellular immune response is even more important that also drives maturation of the humoral immune response. The Cell publication from “Garcia-Beltran et al, 2021” also addressed the presence of neutralization antibodies against SARS-COV-2 after vaccination and their importance in the clinical settings: “While the clinical impact of neutralization resistance remains uncertain, these results highlight the potential for variants to escape from neutralizing…”. These authors of the Cell publication did not overstate their findings. These research scientists also admitted that overall the T-cell immune responses previously triggered by vaccination or infection of one virus variant in contrary to the humoral responses still function to clear other virus variants. This means in the clinical setting, virus variants do not have the disease impact that is being propagated. This opens up the discussion of what so far was ignored in favor of the adaptive immune responses that can be altered by vaccination. Researchers in general have failed to consider the extraordinary role of the native immune system in the respiratory tract: it is dependent on the MALT and BALT immune responses of unspecific IgG and IgA and atypical T-cell responses and other innate immune responses. Tolerance toward foreign antigens is a very important factor for the respiratory tract and this would be jeopardized by a highly aggressive adaptive immune response.

As I stated in my previous review, I would like the authors to compare in the discussion what are the pro and contra of their vaccination strategy to the used mRNA and adenovirus based vaccinations, e.g. the presented vaccination outcome does not have the same problem compared to mRNA vaccinations where mRNA from vaccinations can be found in germinal centers potentially disturbing the general adaptive immune responses (Cell, Lederer et al., 2022). Another comparison with mRNA and Adenovector based vaccinations is the  uncontrollable concentrations of the spike protein expression leading potentially to toxicological problems (see before and Cell, Röltgen et al., 2022). With the authors´ vaccination strategy, it is unknown how long the mRNA or adenovector produces active spike protein, which cells are in vivo transfected and how much and how long the spike protein is expressed and stable in different places of the vaccinated individual body. The authors have the advantage to know how much active substance concentration is injected in an individual.

The authors also conveniently ignore the fact that nothing can be "proven" in natural sciences except in the field of Mathematics. Our research in natural sciences is based on evidences that is carefully balanced by the researcher to achieve the best explanation of the observed evidences. This is what I miss in this manuscript.

Reviewer 2 Report

This reviewer is satisfied with the revisions that have been made to the manuscript. The manuscript can therefore be considered for publication as it is in Vaccines, but should be proofread for minor typographical errors.

Author Response

Thanks for your review, we have proofread the text for minor typographical errors.

Round 3

Reviewer 1 Report

-